# Study on Starch-Based Thickeners in Chyme for Dysphagia Use

**DOI:** 10.3390/foods14010026

**Published:** 2024-12-25

**Authors:** Youdong Li, Lingying Li, Guoyan Liu, Li Liang, Xiaofang Liu, Jixian Zhang, Chaoting Wen, Xin Xu

**Affiliations:** 1College of Food Science and Engineering, Yangzhou University, Yangzhou 225127, China; youdongli@yzu.edu.cn (Y.L.); lilingying@163.com (L.L.); yzufsff@163.com (G.L.); liangli0508@hotmail.com (L.L.); zjx@yzu.edu.cn (J.Z.); chaoting@yzu.edu.cn (C.W.); 2School of Tourism and Cuisine, Yangzhou University, Yangzhou 225127, China; liuxf@yzu.edu.cn

**Keywords:** modified starch, food for special medical purposes, dysphagia, textural, swallowing simulation

## Abstract

A dysphagia diet is a special dietary programme. The development and design of foods for dysphagia should consider both swallowing safety and food nutritional quality. In this study, we investigated the rheological properties (viscosity, thixotropy, and viscoelasticity), textural properties, and swallowing behaviour of commercially available natural, pregelatinised, acetylated, and phosphorylated maize starch and tapioca starch. The results showed that all the samples belonged to food grade 3 in the framework of the International Dysphagia Dietary Standardization Initiative (IDDSI) and exhibited shear-thinning behaviour in favour of dysphagia patients, except for the sample containing pregelatinised starch, which was grade 2. Rheological tests showed that the samples had good structural recovery properties. At the same starch concentration, the elastic modulus of phosphorylated cassava starch FSMP was significantly greater than that of the starch solution, whereas that of acetylated starch was significantly less than that of the starch solution, and the combination of acetylated starch and protein led to a significant viscosity reduction phenomenon, resulting in FSMPs with good stability and fluidity; this may provide an opportunity for the incorporation of more high-energy substructures. The textural results showed that all the samples possessed textural properties of low hardness, low adhesion, and high cohesion, all of which could be used as food for dysphagia patients. This study may provide a theoretical basis for the creation and design of novel nutritional foods for dysphagia.

## 1. Introduction

Dysphagia is a medical condition characterized by an impairment in some aspect of the swallowing mechanism [1]. This impairment can result in coughing or choking, as the abnormal delay in the movement of food boluses during swallowing disrupts the initiation of the swallowing reflex, causing some food to remain in the mouth. Consequently, individuals with dysphagia may experience decreased food intake, leading to weight loss and nutritional deficiencies [2,3]. Global data show that dysphagia can occur at any age, but the risk is higher above the age of 65 years. The prevalence is 13% in the age group 60–70 years, 16% in the age group 70–79 years, and about 33% in the group ≥ 80 years [4,5]. With the challenges posed by population ageing, there is a need to provide more age-oriented products to serve the elderly and improve their quality of life and well-being. Although texture-modified diets are an effective means of treatment, they can cause several complications for patients. Generally, foods that are soft after cooking or mashing are visually unattractive, and their flavour and sometimes nutritional value are diluted, so hospitals and nursing homes are now catering to the special needs of older persons with swallowing difficulties through the use of thickeners.

Food thickeners, including starch and gel-based thickeners, have been widely used in the treatment and care of patients with dysphagia to effectively reduce the risk of aspiration. Although previous studies have focused on gel-based thickeners, such as xanthan gum (XG) and guar gum (GG), starch has attracted the interest of a wide range of researchers due to its unique high concentration and low viscosity [6,7]. However, natural starch has deficiencies such as low emulsifying ability, poor water solubility, and poor stability, which largely limit its use in food for patients with dysphagia. Therefore, the modification of natural starch is an effective way to improve its performance. Currently, modification methods broadly include chemical, physical, biological (enzyme), and composite techniques. Recently, the European Food Safety Authority (EFSA) conducted a comprehensive re-evaluation of twelve modified starches, including oxidised starch, monostarch phosphate, distarch phosphate, phosphated distarch phosphate, acetylated distarch phosphate, acetylated starch, acetylated distarch adipate, hydroxypropyl starch, hydroxypropyl distarch phosphate, starch sodium octenyl succinate, acetylated oxidised starch, and starch aluminium octenyl succinate [8]. These modified starches are authorized for use as food additives according to European Union Food Additives Regulation (EC). No 1333/2008. Rheological properties of commercial edible thickening formulations with gum-based composites for dysphagia care were reported; wherein, the commercial samples selected for the study contained the same main ingredient, i.e., pectin [9]. However, pectin thickeners are only suitable for thickening liquid fluids. Some subsequent studies on thickeners have generated interest in starch-based thickeners. Starches such as corn, potato, and cassava have been studied as food thickeners [3,10]. Moreover, Tsubokawa et al. [11] reported that jelly-like foods made from high amylose rice flour have a texture suitable for the diets of patients with dysphagia. In another study, Yang et al. [12] investigated the formulations and stability of dysphagia-friendly food matrices with calorie-dense starchy thickeners. Kim, Jeong, and Yoo [13] studied the flow behaviour of thickened water samples prepared with six starch-based food thickeners.

Currently, pregelatinised starches, acetylated starches, and phosphorylated starches are available on the market. However, despite the many reports on the properties of these starches, there is little literature on their structure, functional properties, and thickening properties. More importantly, their effects and mechanisms as thickening ingredients in foods for special medical purposes (FSMP) are not well understood. Therefore, the aim of this study was to investigate the feasibility of commercially available natural pregelatinised starch, acetylated starch, phosphorylated maize starch, and tapioca starch as FSMP thickeners. The study assessed the stability and the rheological and textural properties of the samples, as well as the oral processing behaviour and associated physicochemical behaviour. This study reveals the potential of modified starches for FSMP applications and provides a reference for their application.

## 2. Materials and Methods

### 2.1. Materials

This study utilized the following materials: unmodified starch from corn and cassava (native or without degree of substitution) was purchased from Suguo Supermarket (Yangzhou, China). Pregelatinised starch (degree of gelatinization 100%), acetylated starch (degree of substitution 0.019), and phosphorylated starch from corn (degree of substitution 0.031) were purchased from Xinerrui Biotechnology Co. (Zhengzhou, China). Pregelatinised starch (degree of gelatinization 100%), acetylated starch (degree of substitution 0.024), and phosphorylated starch from cassava (degree of substitution 0.047) were purchased from Dongguan Dongmei Food Co. (Dongguan, China). Soybean isolate protein was purchased from Solarbio (Beijing, China). Nile red, fluorescein isothiocyanate (FITC), and 8-anilino–1-naphthalenesulfonic acid (ANS) were purchased from Sigma-Aldrich (St. Louis, MO, USA). Deionized water was used, and all reagents were commercially available and analytical grade.

### 2.2. Properties of Starch

#### 2.2.1. Scanning Electron Microscopy (SEM)

Morphological observations of the samples were performed using an S-4800 scanning electron microscope (SEM) from Hitachi (Tokyo, Japan) at an accelerating voltage of 2 kV. The samples underwent freezing at −80 °C for 12 h, followed by freeze-drying for 48 h. They were then mounted onto the sample stage using conductive adhesive. After a gold sputtering process, the samples were examined and imaged.

#### 2.2.2. X-Ray Diffraction (XRD)

The crystalline structure of starch was evaluated using X-ray diffraction analysis with a D8 Advance diffractometer (Bruker, Germany). Prior to analysis, all samples were preconditioned to a consistent moisture content to ensure reliable results. The diffraction patterns were obtained in a 2θ range of 5° to 40°, with a step size of 0.02° (2θ) and an X-ray source of Cu–Kα radiation. The operating conditions for the diffractometer were set to 40 kV and 40 mA for optimal analysis [14].

To quantify the crystalline structure of the starch samples, the MDI Jade 6.0 software was utilized, which facilitated the calculation of relative crystallinity (RC). This calculation was performed by determining the ratio of the area attributable to the crystalline fraction to the total diffraction area observed in the X-ray patterns. This method provides a valuable insight into the crystallinity of starch, which is crucial for understanding its functional properties and applications.

#### 2.2.3. Attenuated Total Reflectance–Fourier Transform Infrared (ATR–FTIR) Spectroscopy

The ATR-FTIR spectra of both native and modified starches were obtained using a Cary 610/670 instrument (Palo Alto, CA, USA) following the protocol outlined by Dong and Vasanthan with minor adjustments. In this analysis, dry starch powder was utilized for spectral evaluation, and the ATR-FTIR spectra were recorded across a wavelength range of 4000 to 400 cm^−1^. Each generated spectrum represents the average of 32 individual scans, which were subsequently processed through baseline correction and normalization to enhance data accuracy [15].

For peak resolution enhancement, a gamma factor of 18 and a smoothing factor of 0.11 were applied using Origin 2019 software (Origin Lab, Northampton, MA, USA). The absorbance intensities of the spectral bands observed at approximately 1047, 1035, and 1022 cm^−1^ were specifically analysed, as these bands provide insights into the crystalline structures of both native and modified starch samples. This detailed spectral analysis serves to elucidate the structural differences imparted by the modification processes.

### 2.3. The Properties of FSMP Chyme

#### 2.3.1. Sample Preparation

A 5% starch solution (47.5 g of pure water and 2.5 g of sample powder) was prepared at room temperature (25 °C) and uniformly combined. With continuous stirring, the solution was then placed in a water bath at 95 °C for 15 min to dissolve fully. The solution was then cooled for 1 h to achieve steady-state viscosity.

To prepare the FSMP samples, we utilized a modified formula based on the composition reported by Wei et al. [16]. Initially, starch solutions were prepared. Subsequently, calculated amounts of pea protein (4% *w*/*w*) and soybean oil (4% *w*/*w*) were incorporated into the solutions and stirred at room temperature for one hour. The mixtures were then homogenized using a high shear homogenizer (FLUKO FA25, Essen, Germany) at a speed of 10,000 rpm for five minutes to obtain homogeneous FSMP samples.

#### 2.3.2. Malvern Particle Size Analyzer

The particle size distribution and zeta potential measurements were based on Lin et al. (2018) and Xiong et al. [17,18]. Specifically, the FSMP powder was diluted 500-fold and analyzed for zeta potential using a particle analyser (Zetasizer Nano ZS, Malvern Panalytical, Malvern, UK). The particle size distribution of the minced FSMP samples was obtained using a laser diffraction particle analyser (MasterSizer Hydro 3000MU, Malvern Panalytical, Malvern, UK). The refractive indices of water and starch were 1.330 and 1.652, respectively. All experiments and measurements were conducted in triplicate.

#### 2.3.3. Scanning Electron Microscopy (SEM)

After freeze-drying, the microstructure of the FSMP chyme was analyzed using a field emission scanning electron microscope (S-4800, Hitachi, Tokyo, Japan).

#### 2.3.4. Confocal Laser Scanning Microscope (CLSM)

The microstructural analysis of the FSMP chyme was conducted utilizing a Zeiss LSM 700 confocal laser scanning microscope (CLSM), manufactured by Carl Zeiss Microlmaging GmbH in Jena, Germany, with a magnification factor of 20×, following the protocols outlined by Drapala et al. [19,20]. For the staining process, lipids were tagged with Nile red, prepared at a concentration of 0.25 mg/mL in DMSO, and subsequently diluted to 1 μg/mL. Additionally, starch was labelled with FITC, dissolved in DMSO at a concentration of 1 mg/mL, and further diluted to 10 μg/mL.

#### 2.3.5. Rheological Characterization

Rheological properties were assessed using a rheometer (AR-2000, TA Instruments, Newcastle, UK) following the method outlined by Du et al. [21], with minor modifications. Select a 40 mm fixture with a gap set to 1000 μm. Dynamic stress scanning experiments were conducted at a constant frequency of 1 Hz, varying stress levels from 0.1 to 1000 Pa to identify the linear viscoelastic region. Steady shear tests spanned a shear rate range of 0.1 to 100 s^−1^. Additionally, a time scan was performed to evaluate viscosity variations under alternating low and high shear rates, specifically applying 0.1 s^−1^ for 500 s, followed by 10 s^−1^ for 300 s, and returning to 0.1 s^−1^ for another 500 s. This approach aimed to investigate the thixotropic recovery capability. All experiments were carried out in triplicate at a controlled temperature of 25 °C.

#### 2.3.6. Texture Properties

The texture properties of the samples were rigorously examined using a texture analyser (TWS-Pro, Leicestershire, UK), following a modified methodology based on the protocols described by Liu et al. [22], All samples underwent a comprehensive analysis in texture profile analysis (TPA) mode. A P/38 R mm probe was utilized for this purpose, with the trigger force carefully set at 0.5 N to ensure precision. The analysis encompassed pre-test, test, and post-test speeds of 1 mm/s, alongside a specific compression distance tailored to the dimensions of the samples.

#### 2.3.7. Swallowing Simulation

Using Hadde et al.’s [23] method, swallowing was simulated using a textural analyser (TWS-Pro, TA, UK). In this study, the closest rate to the human swallowing rate of 0.2 mm/s (*v*_1_) was chosen to simulate swallowing work and pushing force. Using the principle that the volume of a probe immersed in a sample in t seconds’ time is the same as the volume of the sample extruded, a required plate probe propulsion rate of 0.34 mm/s (*v*′_1_) was obtained.
$${v^{\prime }}_{1}t\pi {\left(\frac{d_{1}}{2}\right)}^{2}=v_{1}t\pi \left[{\left(\frac{D_{1}}{2}\right)}^{2}-{\left(\frac{d_{1}}{2}\right)}^{2}\right]$$

For the test, approximately 50 mL of a sample was placed into a sample vial (the height of the solution to be tested was greater than or equal to 75% of the height of the sample cylinder). The sample was tested using a textural analyser to simulate swallowing (diameter of the plate probe (d_1_): 38 mm; diameter inside the D_1_ sample cylinder (D_1_): 42 mm).

At room temperature (25 °C), a single compression mode was applied to test the probe; a 0.5 N trigger force was applied to it, and it continued to advance 10 mm. The pre-test and mid-test speeds were the same at 0.34 mm/s, and the post-test speed was 1 mm/s. The measured work done by compression was the work required for safe swallowing.

#### 2.3.8. IDDSI (International Dysphagia Diet Standardisation Initiative) Tests

The IDDSI framework categorizes the foods and drinks that patients with dysphagia can consume into levels that describe food texture and drink thickness. IDDSI testing methods evaluate the flow of a liquid or textural characteristics of a solid to determine its categorization [23]. In this study, the 10 mL syringe experiment (the inner diameter of the syringe barrel and the outlet tip are 15 mm and 0.7 mm, respectively, predominantly used to measure drink thickness) was applied. A syringe was held vertically with its tip blocked. A sample solution was injected into the syringe to the 10 mL mark. Then, the syringe tip was unblocked and, after 10 s, the remaining volume of the sample solution in the syringe was observed.

### 2.4. Statistical Analysis

All data presented in this paper were derived from the standard deviation (SD) calculations of results obtained from three parallel experiments. Statistical analyses were conducted using one-way ANOVA followed by Duncan’s test, with a significance level set at *p* < 0.05. Data analysis was performed utilizing SPSS version 22.0, while plots and further calculations were generated using Origin 2019 and Jade 6.0.

## 3. Results and Discussion

### 3.1. Properties of Starch

#### 3.1.1. Morphology of Starch Granules

Different chemical, enzymatic, and physical processes can induce structural modifications in starch, which can be detected through microscopic analysis [24]. The microstructures of the natural and modified corn and cassava starch samples were observed using scanning electron microscopy (SEM). The surface and shape characteristics of the starch samples are shown in Figure 1. Natural corn starch granules have solid spheres and polygonal morphology, while natural cassava starch granules have round, oval, or oval truncated shapes. The SEM images show that the surface of the natural starch granules is smooth and without cracks or pores. However, the granular structure of the pregelatinised starch has been completely lost, showing an irregular, lamellar structure, which is in line with previous reports [25].

Acetylation induced the formation of distinct concave pits on the surfaces of starch granules, resulting in an augmentation of their molecular aggregation and porosity. Furthermore, the acetylated starches exhibited a notable degree of granule fusion, which is attributed to the incorporation of carbonyl groups [26]. These carbonyl groups facilitated enhanced hydrogen bonding and non-covalent interactions among the polymeric chains, thereby contributing to the observed granule fusion [27]. Moreover, phosphorylated starch has been found to have similar morphological characteristics as those of native starch. Bidzińska et al. found similar changes in phosphorylated potato starch granules [28].

#### 3.1.2. Structural Characteristics of Starches

The crystal type and content of starch can be determined by XRD, the results of which are shown in Figure 2A. The characteristic diffraction peaks of native, acetylated, and phosphorylated starches starch appeared at 2θ: 15, 17, 18, and 23°, corresponding to the A-type polymorph, which was in good accordance with previous research [29]. Compared to native starch, the crystal structure of pregelatinized starch was completely destroyed. The heat treatment broke intra- and intermolecular hydrogen bonds in the starch granules, resulting in the disruption of the regularity of the crystal structure [25]. The peaks of acetylated and phosphorylated starch showed only slight changes, indicating that acetylation and phosphorylation had little effect on the crystalline forms of corn and cassava starch. Therefore, we concluded that acetylation and phosphorylation mainly occur in the amorphous region of starch granules and have no effect on the crystallization pattern.

The crystallinity of starch was calculated using the area method. The results showed that the crystallinity of phosphorylated starch was significantly higher (Table 1). The increase in the crystallinity of phosphorylated cassava starch was larger than that of phosphorylated corn starch, as phosphorylation occurs mainly on branched chains in starch, and there are fewer in corn starch. However, the relative crystallinity of acetylated starch was very similar to that of natural starch. Kim et al. [30] and Luo and Shi [31] also reported no difference in XRD patterns or relative crystallinity between natural and acetylated corn, yellow pea, cowpea, and chickpea starches.

The ATR-FTIR technique was used to study the surface structural characteristics of the starch granules. The spectra of the natural and modified starches are shown in Figure 2B. The ATR-FTIR spectra of natural corn starch and modified starch were almost identical, while variations in the spectra of cassava starch occurred between 3337–3240 cm^−1^. This was attributed to –OH stretching [32]; the breadth indicates the degree of inter- and intramolecular hydrogen formation. While cassava has highly branched starch properties compared to corn starch, starch modification results in the stretching and changing of intermolecular bonds.

### 3.2. Properties of the FSMP Samples

#### 3.2.1. Particle Size Distribution and Zeta Potential

Figure 3A shows the particle size distribution of the minced FSMP samples prepared from different starches. As shown in the figure, except for the acetylated starch (which had two similar peaks), the particle size distribution of the minced food made from corn starch was the most consistent, with one main peak and one smaller secondary peak. In contrast, the secondary peaks of the pre-gelatinized starch were smaller, indicating a more uniform particle size distribution in the FSMP chyme. The main peaks of acetylated cassava starch were larger, and there was a more uniform particle size distribution compared to the other starches. From the figure, it can be seen that the particle size distribution of acetylated cassava starch is more homogeneous and has relatively better stability.

Figure 3B depicts the zeta potential values of FSMP samples formulated using diverse starches. Zeta potential serves as a crucial indicator of the electrical surface characteristics of colloidal particles suspended in a liquid, reflecting the degree of electrical attraction and repulsion between suspended solids’ surfaces based on their potential values. Higher zeta potential values signify increased stability and a reduced tendency for particles to aggregate, indicating that the particles are stable and free from aggregation.

Particle size and zeta potential indicate the stability of the system, and a stable chyme is more suitable for patients with dysphagia.

#### 3.2.2. Microscopic Morphology (SEM, CLSM)

Figure 4A,B show SEM and CLSM photographs of the FSMP samples with different starch additions. The results showed an irregular starch and protein matrix with small globules representing fat embedded in chyme network. Similar microstructures were also observed by Ahsan, et al. [33] and Hu, et al. [34]. It can be clearly seen in Figure 4 that starches and proteins were bound together in different ways. Clear fat particles can be seen on the surface of the electron micrographs of the pre-gelatinized starch FSMP samples. This may be due to the disruption of the crystal morphology of the pregelatinised starch as well as the starch’s wrapping around the surface of the protein particles to form a stable spherical particle (a result verified by laser confocal microscopy). The acetylated and phosphorylated starch FSMP samples had more homogeneous surfaces with only a few spherical granules, which may be related to the chemical modification that introduced the acetyl and phosphate groups, allowing the formation of a complex network infrastructure between starches and proteins.

#### 3.2.3. Rheological Properties and Textural Properties

The coordination between the rheological and textural properties of fluid food, the propulsive forces exerted by the oropharyngeal muscular system, and the biomechanical factors protecting the respiratory tract determine a safe swallowing process. The rheology of fluids achieving this coordination is considered the most important aspect of dysphagia management.

##### Shear Viscosity

Figure 5 shows the flow curves of the starch solutions and the FSMP samples. As shown, a common feature of both the starch solutions and FSMP samples is pseudoplasticity, which is required in dysphagia treatment. Pregelatinized starches have poor shear dependence compared to other starches and cannot maintain viscosity within the desired ranges. The viscosity of the corn starch FSMP samples was slightly lower than that of the aqueous starch solution with the same solubility and was without much variation, which may be related to the high amylose content of corn starch. The viscosity of corn starch is provided mainly by its amylopectin, as reported by Kankate et al. [35]. The viscosity of the phosphorylated cassava starch FSMP was significantly higher than that of the aqueous solution with the same starch concentration. This may be due to the introduction of hydrophilic phosphate groups, resulting in a significant increase in the viscosity of the phosphorylated starch samples. The viscosity of acetylated cassava starch FSMP was significantly reduced, probably due to the spatial repulsion and hydrophobic effects of acetyl groups that hinder starch–protein structural association.

To enhance our understanding of the shear-thinning behaviour of polysaccharide solutions and the thickening of emulsions, the Ostwald–de Waele model was employed in this study.
η = Kγ^n−1^(1)

The apparent shear viscosity (η) is defined in pascal-seconds (Pa·s), while the shear rate (γ) is measured in inverse seconds (s^−1^). The consistency index (K) is expressed in pascals raised to the power of the flow index (Pa·sn), with ’n’ indicating the flow behaviour of the fluid. In the case of Newtonian fluids, n equals 1. For pseudoplastic fluids, the value of n falls between 0 and 1, with deviations from 1 signifying the degree of non-Newtonian characteristics exhibited by the fluid. Table 2 presents the parameters of starch solutions in the FSMP samples analysed using the power law model. The n exponents fall within the range of 0 to 1, indicating that all the samples behave as pseudoplastic fluids. Notably, the n values for the FSMP samples were significantly higher than those of the corresponding starch solutions, suggesting pronounced Newtonian characteristics. This non-Newtonian behaviour permits the neuromuscular system to maintain a longer reflex response time for closing the epiglottis, which aids in swallowing and reduces the risk of aspiration [37].

Furthermore, an estimate of *η*_10_, *η*_50_, and *η*_100_, the viscosity at the shear rate 10 s^−1^, 50 s^−1^, and 100 s^−1^, separately, was extracted from Equation (1) since they are generally considered as the most representative values for the normal swallowing process. This unexpected viscosity property of starch in chyme can be translated as an advantage instead, allowing more addition of starch as energy substance [38].

##### Thixotropic Recovery Capacity

Thixotropic reversion properties have a major impact on food intended for patients with dysphagia. Therefore, the samples were subjected to a three-stage thixotropic test, as shown in Figure 6. At a low shear rate of 0.1 s^−1^, the samples’ viscosities did not change significantly due to certain rheological resistance properties [39]. However, at a high shear rate of 10 s^−1^, the viscosities of all samples decreased significantly, exhibiting shear thinning properties. The acetylated and phosphorylated starches in the FSMP samples decreased more, displaying higher shear sensitivity and greater shear thinning ability. The viscosity increased sharply when the shear rate returned to 0.1 s^−1^, showing good structural recovery properties.

##### Viscoelastic Properties

Although most studies on food for patients with dysphagia have focused on viscosity, viscoelasticity is also important in the overall effect of safe swallowing. To investigate viscoelasticity, storage modulus (G′), loss modulus (G″), and mechanical loss (tan δ), we performed amplitude oscillatory stress scans at a constant frequency of 1 Hz using stresses in the range of 0.1–100 Pa. As shown in Figure 7, the pre-dextrinized starch solution and its FSMP sample were viscous throughout the tested concentration range. In contrast, the other starches and their FSMP samples exhibited similar solid behaviour in the linear viscoelastic region of G′ > G″ (tan δ < 1). At the same concentrations, acetylated corn starch and phosphorylated starches showed similar viscoelasticity, which may be related to their high amylose content. Consistent with the results of the previous flow scans, the elastic modulus of the phosphorylated cassava starch FSMP was significantly larger than that of the starch solutions at the same concentrations, while the elastic modulus of acetylated starch was significantly smaller.

##### Textural Properties

Texture, as a sensory attribute, plays a pivotal role in evaluating the palatability and swallowing ease of food products [40]. The overall texture of the FSMP was meticulously assessed using the TPA (Texture Profile Analysis) model, as presented in Table 3. Adhesiveness, a key parameter within this model, quantifies the adhesive force between the probe and the gel sample, thereby providing insights into the adhesion between food and teeth. Foods exhibiting high adhesiveness may not be ideal for individuals seeking a smooth and effortless swallowing experience [41]. In the present study, adhesion was measured to predict the state of the FSMP at the pharynx and mouth. With less adhesion, less food residue will be left near the pharynx after swallowing, thereby protecting it and facilitating eating [42]. Overall, the adhesiveness values were kept at relatively low levels for all the FSMP samples tested, which was needed to decrease the risk of pharyngeal residue. This suggests that starch-containing FSMP samples easily detach from the teeth/palate and are suitable for consumption as food for patients with dysphagia.

The hardness of a semisolid food is indicative of the energy required for its rupture during mastication. Low hardness and less adhesive properties make biting and squeezing between the tongue and hard palate easier, which reduces the energy needed to swallow.

Cohesiveness signifies the internal binding force within the gel network, underlining the capacity to withstand the breakdown of food into smaller particles during chewing. Adequate cohesiveness is indispensable for any diet designed for dysphagia patients [40]. Notably, the overall adhesiveness values for all the tested FSMP samples remained comparatively lower than those reported by Dick et al. [43], suggesting an improvement in texture that favours ease of swallowing.

### 3.3. Swallowing Simulation

The swallowing work was simulated using the plate counter-squeeze mode of the textural analyser. As shown in Figure 8A, more work was required to swallow the FSMP samples containing acetylated corn starch, meaning that the neuromuscular system would have a longer response time when closing the epiglottis, facilitating consumption.

The syringe flow test suggested by the IDDSI, as previously mentioned, has five levels [28]:Level 0 (thin): <1 mL remaining in the syringeLevel 1 (slightly thick): 1–4 mL remaining in the syringeLevel 2 (mildly thick): 4–8 mL remaining in the syringeLevel 3 (moderately thick): >8 mL remaining in the syringeLevel 4 (extremely thick): 10 mL (no liquid flow)

As shown in Figure 8B, all the minced foods were Level 3, except for those containing pregelatinised starch, which were Level 2.

## 4. Conclusions

The study investigated the feasibility of commercially available natural and pregelatinised starches from maize and cassava, acetylated and phosphorylated, as thickening agents for special medical food chyme. The rheological results showed that the chyme with thickening agents was a pseudoplastic fluid with good shear thinning ability, good thixotropic recovery, suitable viscoelasticity, and significant viscosity reduction in combination with protein, which is expected to be a new type of thickening agent for dysphagia. Pectin thickeners are commonly used in liquid beverages (water, juice, milk, etc.), but patients with dysphagia need whole-nutrient chyme to help them recover, and our thickeners provide an advantage for adding more energetic substances to swallowed food formulations. This study will provide a theoretical basis for the development of dysphagia foods.

## Figures and Tables

**Figure 1 foods-14-00026-f001:**
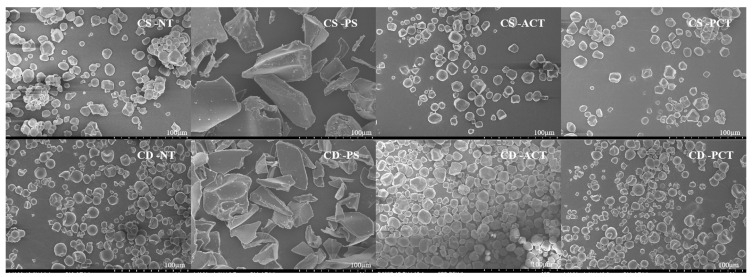
SEM of natural and modified starch. (CS-NT: corn ordinary starch, CS-PS: corn pregelatinised starch, CS-ACT: corn acetylated starch, CS-PCT: corn phosphorylated starch, CD-NT: cassava ordinary starch, CD-PS: cassava pregelatinized starch, CD-ACT: cassava acetylated starch, CD-PCT: cassava phosphorylated starch).

**Figure 2 foods-14-00026-f002:**
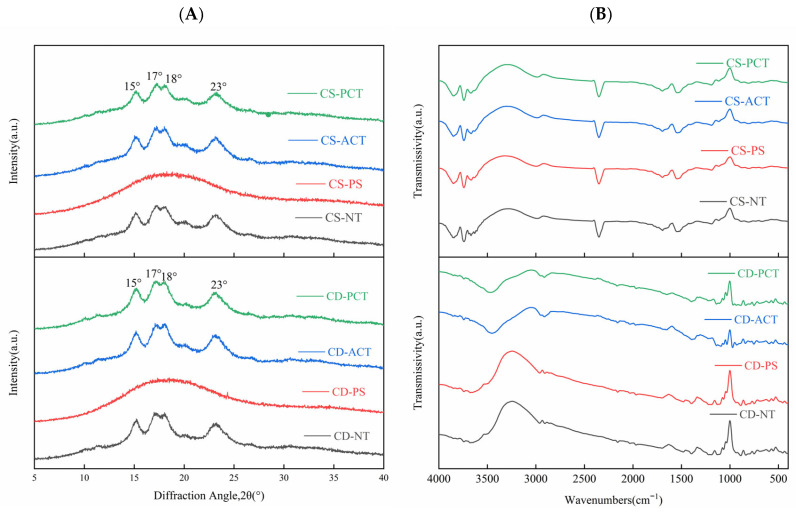
(**A**) XRD and (**B**) FITR of natural and modified starch.

**Figure 3 foods-14-00026-f003:**
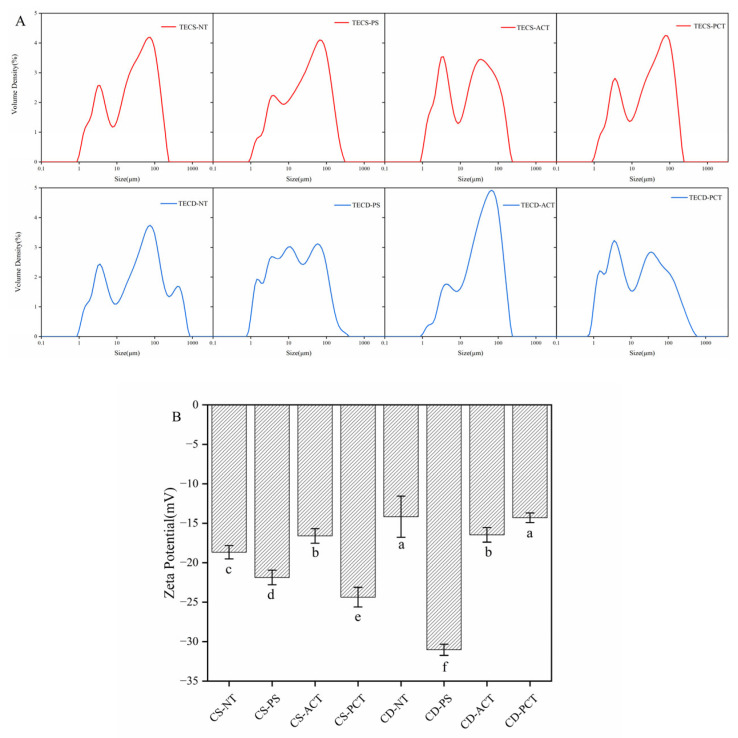
(**A**) Particle size distribution of FSMP samples with different types of starch. (**B**) The effect of different types of starch on the zeta potential of FSMP samples. (TECS-NT: FSMP samples containing native corn starch, TECS-PS: FSMP samples containing pregelatinized corn starch, TECS-ACT: FSMP samples containing acetylated corn starch, TECS-PCT: FSMP samples containing phosphorylated corn starch, TECD-NT: FSMP samples containing native cassava starch, TECD-PS: FSMP samples containing cassava pregelatinized starch, TECD-ACT: FSMP samples containing cassava acetylated starch, TECD-PCT: FSMP samples containing cassava phosphorylated starch; The different letters (a–f) represent significant differences between the groups).

**Figure 4 foods-14-00026-f004:**
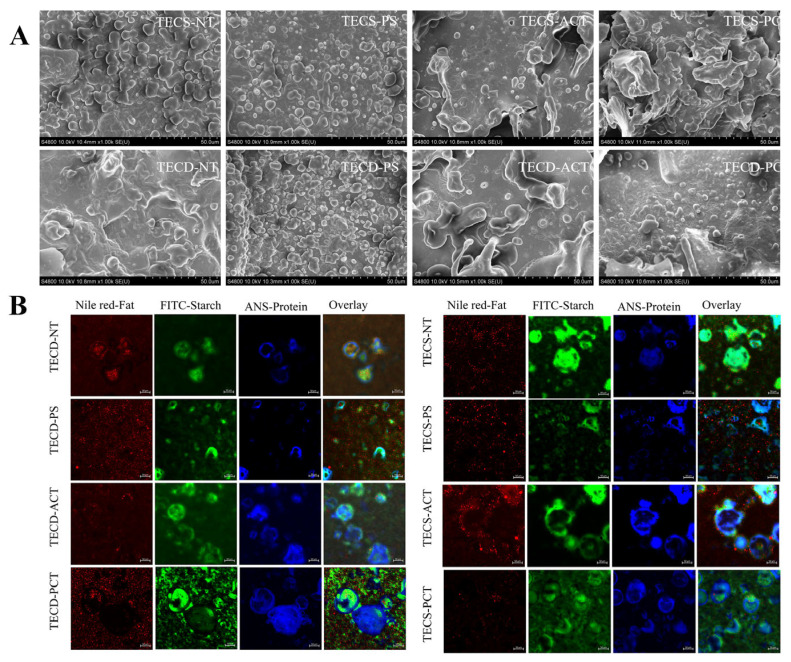
(**A**) SEM of FSMP samples with different types of starch. (**B**) Confocal Laser Scanning Microscopy images of FSMP samples with different types of starch. (The first column Nile red labelled fat, second column FITC labelled starch, third column ANS labelled Protein, last column overlay image with fat in red, starch in green and protein in blue. Each row represents one sample. Scale bar 50 μm). (TECS-NT: FSMP samples containing native corn starch, TECS-PS: FSMP samples containing pregelatinized corn starch, TECS-ACT: FSMP samples containing acetylated corn starch, TECS-PCT: FSMP samples containing phosphorylated corn starch, TECD-NT: FSMP samples containing native cassava starch, TECD-PS: FSMP samples containing cassava pregelatinized starch, TECD-ACT: FSMP samples containing cassava acetylated starch, TECD-PCT: FSMP samples containing cassava phosphorylated starch).

**Figure 5 foods-14-00026-f005:**
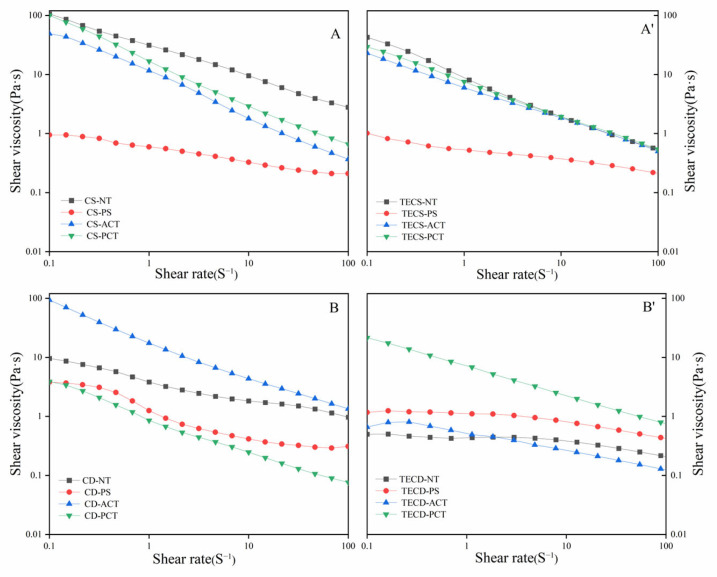
Steady shear flow curves of FSMP samples [36]. (**A**): Corn starch solution, (**B**): Cassava starch solution, (**A’**):Corn starch special medical chyme, (**B’**): Cassava starch special medical chyme’ (TECS-NT: FSMP samples containing native corn starch, TECS-PS: FSMP samples containing pregelatinized corn starch, TECS-ACT: FSMP samples containing acetylated corn starch, TECS-PCT: FSMP samples containing phosphorylated corn starch, TECD-NT: FSMP samples containing native cassava starch, TECD-PS: FSMP samples containing cassava pregelatinized starch, TECD-ACT: FSMP samples containing cassava acetylated starch, TECD-PCT: FSMP samples containing cassava phosphorylated starch).

**Figure 6 foods-14-00026-f006:**
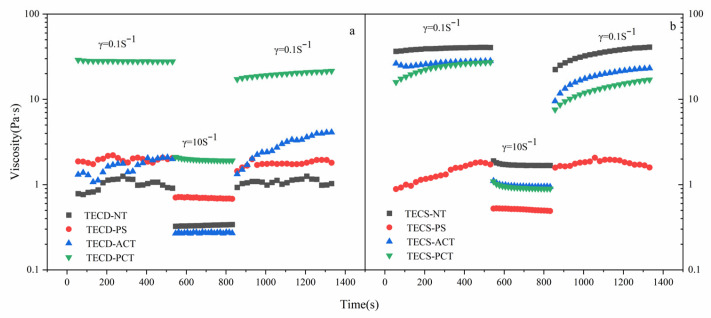
Thixotropic recovery curves of FSMP samples [34] ((**a**): Corn starch special medical chyme, (**b**): Cassava starch special medical chyme’) (TECS-NT: FSMP samples containing native corn 1starch, TECS-PS: FSMP samples containing pregelatinized corn starch, TECS-ACT: FSMP samples containing acetylated corn starch, TECS-PCT: FSMP samples containing phosphorylated corn starch, TECD-NT: FSMP samples containing native cassava starch, TECD-PS: FSMP samples containing cassava pregelatinized starch, TECD-ACT: FSMP samples containing cassava acetylated starch, TECD-PCT: FSMP samples containing cassava phosphorylated starch).

**Figure 7 foods-14-00026-f007:**
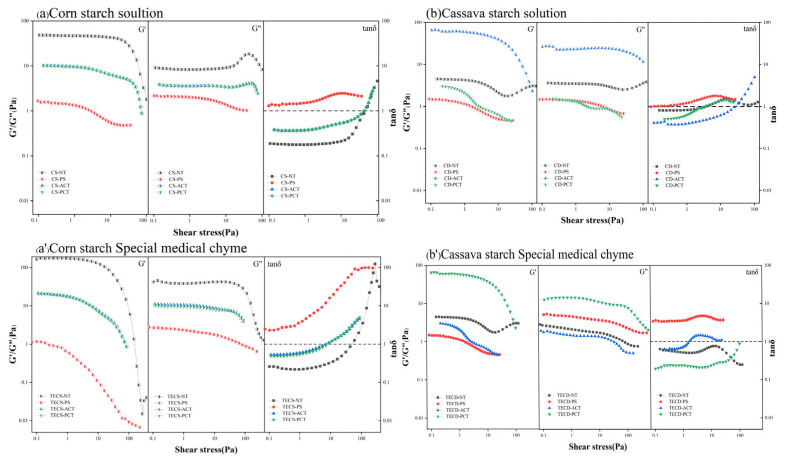
Strain–dependence of dynamic viscoelasticity for different types of starch solutions and special medical chyme. (TECS-NT: FSMP samples containing native corn starch, TECS-PS: FSMP samples containing pregelatinized corn starch, TECS-ACT: FSMP samples containing acetylated corn starch, TECS-PCT: FSMP samples containing phosphorylated corn starch, TECD-NT: FSMP samples containing native cassava starch, TECD-PS: FSMP samples containing cassava pregelatinized starch, TECD-ACT: FSMP samples containing cassava acetylated starch, TECD-PCT: FSMP samples containing cassava phosphorylated starch).

**Figure 8 foods-14-00026-f008:**
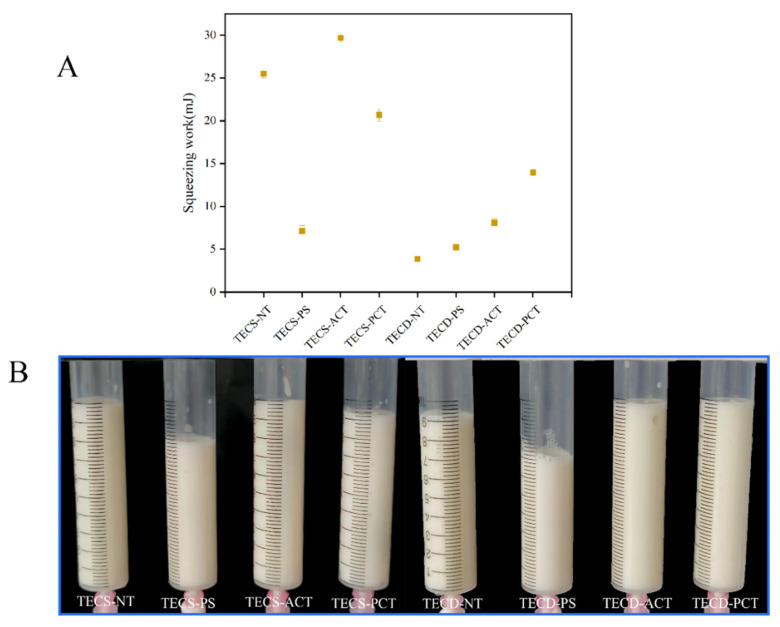
(**A**) Plate shaped reverse extrusion simulation of swallowing with special medical chyme containing different starches. (**B**) IDDSI testing of special medical chyme containing different starches [34]. (TECS-NT: FSMP samples containing native corn starch, TECS-PS: FSMP samples containing pregelatinized corn starch, TECS-ACT: FSMP samples containing acetylated corn starch, TECS-PCT: FSMP samples containing phosphorylated corn starch, TECD-NT: FSMP samples containing native cassava starch, TECD-PS: FSMP samples containing cassava pregelatinized starch, TECD-ACT: FSMP samples containing cassava acetylated starch, TECD-PCT: FSMP samples containing cassava phosphorylated starch).

**Table 1 foods-14-00026-t001:** Degree of substitution and relative crystallinity of native and modified starch.

	CS-NT	CS-PS	CS-ACT	CS-PCT	CD-NT	CD-PS	CD-ACT	CD-PCT
DS (%)	0	100	0.019	0.031	0	100	0.024	0.047
Crystallinity	30.82	0.00	29.94	44.84	23.19	0.00	22.30	52.82

**Table 2 foods-14-00026-t002:** Viscosities at the shear rates of 10 s^−1^, 50 s^−1^, 100 s^−1^, and power law parameters of starch solution and FSMP samples [34]. (TECS-NT: FSMP samples containing native corn starch, TECS-PS: FSMP samples containing pregelatinized corn starch, TECS-ACT: FSMP samples containing acetylated corn starch, TECS-PCT: FSMP samples containing phosphorylated corn starch, TECD-NT: FSMP samples containing native cassava starch, TECD-PS: FSMP samples containing cassava pregelatinized starch, TECD-ACT: FSMP samples containing cassava acetylated starch, TECD-PCT: FSMP samples containing cassava phosphorylated starch).

Starch	*K* (Pa s^n^)	*n*	R^2^	Chyme	*K* (Pa s^n^)	*n*	R^2^
CD-NT	4.176	0.631	0.993	TECD-NT	0.420	0.874	0.850
CD-PS	1.525	0.551	0.942	TECD-PS	1.010	0.852	0.890
CD-ACT	17.602	0.279	0.999	TECD-ACT	0.488	0.754	0.935
CD-PCT	0.975	0.378	0.995	TECD-PCT	7.108	0.515	0.999
CS-NT	20.534	0.295	0.999	TECS-NT	9.591	0.339	0.998
CS-PS	0.608	0.796	0.936	TECS-PS	0.561	0.782	0.981
CS-ACT	12.031	0.362	0.994	TECS-ACT	6.129	0.431	0.999
CS-PCT	17.902	0.247	0.999	TECS-PCT	7.759	0.413	0.999

**Table 3 foods-14-00026-t003:** Textural characteristics of FSMP containing different starches. (TECS-NT: FSMP samples containing native corn starch, TECS-PS: FSMP samples containing pregelatinized corn starch, TECS-ACT: FSMP samples containing acetylated corn starch, TECS-PCT: FSMP samples containing phosphorylated corn starch, TECD-NT: FSMP samples containing native cassava starch, TECD-PS: FSMP samples containing cassava pregelatinized starch, TECD-ACT: FSMP samples containing cassava acetylated starch, TECD-PCT: FSMP samples containing cassava phosphorylated starch). (Different letters indicate the presence of significant significant differences (*p* < 0.05)).

	Hardness (N)	Adhesion (mJ)	Elasticity (mm)	Adhesiveness (N)	Chewiness (mJ)	Cohesiveness (Ratio)
TECS-NT	0.301 ± 0.021 ^b^	0.350 ± 0.001 ^bc^	5.690 ± 0.272 ^a^	0.213 ± 0.003 ^b^	1.217 ± 0.058 ^b^	0.710 ± 0.052 ^b^
TECS-PS	0.156 ± 0.015 ^cd^	0.310 ± 0.000 ^bc^	2.817 ± 1.277 ^a^	0.143 ± 0.015 ^de^	0.410 ± 0.195 ^e^	0.920 ± 0.010 ^a^
TECS-ACT	0.356 ± 0.012 ^a^	0.467 ± 0.027 ^a^	6.123 ± 0.181 ^a^	0.252 ± 0.008 ^a^	1.543 ± 0.021 ^a^	0.710 ± 0.046 ^b^
TECS-PCT	0.332 ± 0.053 ^ab^	0.242 ± 0.025 ^c^	4.720 ± 0.594 ^a^	0.222 ± 0.014 ^ab^	1.050 ± 0.140 ^bc^	0.680 ± 0.114 ^b^
TECD-NT	0.192 ± 0.032 ^c^	0.357 ± 0.176 ^ab^	3.990 ± 1.675 ^a^	0.174 ± 0.026 ^cd^	0.687 ± 0.258 ^d^	0.910 ± 0.010 ^a^
TECD-PS	0.132 ± 0.012 ^d^	0.275 ± 0.011 ^bc^	6.157 ± 1.575 ^a^	0.115 ± 0.011 ^ef^	0.697 ± 0.140 ^d^	0.873 ± 0.015 ^a^
TECD-ACT	0.147 ± 0.025 ^cd^	0.085 ± 0.004 ^d^	2.003 ± 0.329 ^a^	0.106 ± 0.026 ^f^	0.207 ± 0.023 ^e^	0.717 ± 0.068 ^b^
TECD-PCT	0.287 ± 0.041 ^b^	0.240 ± 0.063 ^c^	4.500 ± 0.606 ^a^	0.195 ± 0.034 ^bc^	0.867 ± 0.105 ^cd^	0.677 ± 0.068 ^b^

## Data Availability

The original contributions presented in the study are included in the article; further inquiries can be directed to the corresponding author.

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
