# Peer review of "Study on Starch-Based Thickeners in Chyme for Dysphagia Use"

_foods, 2024, doi:10.3390/foods14010026_

Round 1

Reviewer 1 Report

Comments and Suggestions for Authors

Reviewer comments

The authors investigated the rheological, textural properties, and swallowing behavior of different types of starch used as thickeners in Food for special medical purposes. The study found that the combination of acetylated cassava starch and protein resulted in a significant viscosity reduction phenomenon that produced FSMP with good stability and flowability. This study provided some basic knowledge for developing foods for dysphagia patients. The manuscript was well written. However, some minor mistakes need to be revised before it can be published in the Journal of Foods.

Line 22: delete the space before the last sentence.

Line 88: Is “Electronic scanning electron microscopy” correct? Or need to change to “scanning electron microscopy”.

Line 91: change the hyphen sign “-” in “-80 oC” to the end-dash sign ““. The same in line 105 with cm-1 and other places.

Line 182: IDDSI should be written in full form at the first mention.

Line 255: 3.2.1 should be 3.2.1.

Line 338: Delete underline in Eq.(1)

The minus sign of “10 s1” in line 337 and “0.1 s-1" in 350 should be the same. The same with the other places.

Line 389: “low hardness...” should be “Low hardness...”

Line 418: “...medical food chime” should be “medical food chyme.”

- In Table and Figure captions, there should be no periods between “Table” and numbers. For example, “Table. 1.” should be “Table 1.”

The first letters of words in tables should be capitalized. For example, in table 2, “starch” should be “Starch”, ...

- The names of samples are long and hard to remember, which might make the results difficult for readers to understand. Therefore, the authors are recommended to present figures and tables in the same manner as Figure 3.

Author Response

Reviewer #1: 

Comment 1: Line 22: delete the space before the last sentence.

Response 1: Thank you very much for your suggestion, We have made some changes in the manuscript. (Page 1, Line 23)

Comment 2: Line 88: Is “Electronic scanning electron microscopy” correct? Or need to change to “scanning electron microscopy”.

Response 2: Thank you very much for your suggestion, We have made some changes in the manuscript. (Page 2, Line 94)

Comment 3: Line 91:Line 182: IDDSI should be written in full form at the first mention.

Response 3: Thank you very much for your suggestion, We have made some changes in the manuscript. (Page 5, Line 203)

Comment 4: Line 255: 3.2.1 should be 3.2.1.

Response 4: Thank you very much for your suggestion, We have made some changes in the manuscript. (Page 7, Line 279)

Comment 5: Line 338: Delete underline in Eq.(1)

Response 5: Thank you very much for your suggestion, We have made some changes in the manuscript.

Comment 6: The minus sign of “10 s−1” in line 337 and “0.1 s-1" in 350 should be the same. The same with the other places.

Response 6: Thank you very much for your suggestion, We have made some changes in the manuscript.

Comment 7: Line 389: “low hardness...” should be “Low hardness...”

Response 7: Thank you very much for your suggestion, We have made some changes in the manuscript. (Page 13, Line 457)

Comment 8: Line 418: “...medical food chime” should be “medical food chyme.”Response 8:  Thank you very much for your suggestion, We have made some changes in the manuscript.

Comment 9: - In Table and Figure captions, there should be no periods between “Table” and numbers. For example, “Table. 1.” should be “Table 1.”The first letters of words in tables should be capitalized. For example, in table 2, “starch” should be “Starch”, ...

Response 9: Thank you very much for your suggestion, We have made some changes in the manuscript.

Comment 10: The names of samples are long and hard to remember, which might make the results difficult for readers to understand. Therefore, the authors are recommended to present figures and tables in the same manner as Figure 3.

Response 10: Thank you very much for your suggestion. We've made additions.

Reviewer 2 Report

Comments and Suggestions for Authors

The contents of this paper appear like good research as such, and the proper application of several instrumental methods to characterize starch-based formulations for dietary food, for patients with dysphagia.

Generally, the quality of the English is quite good, but I see several grammar or typing errors. For example, a capital at the front of sentence is missing several times, and some sentences have an illogic set up. Probably, a thorough spelling-grammar check could reveal and repair this.  

I have one big problem with this paper, as well as some more detailed comments as listed below per section.

What I truly miss in this paper is which of the starches/FSMP chyme formulations/samples is an already existing/commercially available one. That should be used as a comparison to assess any improvement due to the use of novel ingredients or formulations. But I do not find that anywhere in the paper – but, surprisingly ONLY in the abstract a minor amount of such information is given. It looks like the authors do not realize that an abstract is not an introduction nor a conclusion but a stand-alone text and that the conclusions in it have to be supported by information from the paper itself.

Since this is missing, it is not clear at all what the real aspect of novelty is in this work.

Related to this, I also miss a mention of the demands for the parameters in more detail, such as which Zeta-potential is good enough for the intended food application. In paragraph 3.2.3 and further on in the rheology results, such an assessment is indeed indicated, like what rheological properties are required to ease swallowing. Why is this not made for other test results as well?

In my opinion, adding such information to the main texts of the paper is at least one necessary condition to be met before publication of this manuscript can be considered. So as a recommendation, I must select ‘major revisions.

DETAILED COMMENTS PER SECTION

Introduction

On medical websites, I also see lower figures (like 8% overall) for the incidence of dysphagia. Please mention whether these data are for one certain country or region, or for the worldwide population.

L50 When an abbreviation such as EFSA is used, please inform the readers what this stands for. Same for the statement in lines 55-6which has no meaning without background knowledge.

L66 Same for FSMP, which is written out in the abstract BUT the abstract is stand-alone and NOT an introduction, vide supra.

Most important, the introduction should be extended according to the major issue mentioned above.

Methods

L114 combined with? what?

The phrase pre-dextrinized starch is used several times in the paper but then I would expect to have a description of that in the ‘Materials’ section.

L173 Not any mass spectrometer that I ever met in the laboratory or that I see in the catalogues of lab suppliers can simulate swallowing behavior. Searching on the internet reveals something that is called a MASA test. Is there an unfortunate spelling error, do the authors mean another type of device, not a true mass spectrometer?

 Results

There are no numbers for DS in Table 1. OK, these are mentioned in paragraph 2.1 but that is far away from the results – why not mention the data here as well – much more comfortable for the reader.

L 381 Chocking?? Must be choking probably?

Conclusions

See the main general comment. The conclusions should be extended by an assessment of the novel versus the existing FMSP formulations. By the way, it looks like two times the same sentence with only one term different. This looks bad and can be formulated more logically.

Author Response

Reviewer #2: 

Comment 1: What I truly miss in this paper is which of the starches/FSMP chyme formulations/samples is an already existing/commercially available one. That should be used as a comparison to assess any improvement due to the use of novel ingredients or formulations. But I do not find that anywhere in the paper – but, surprisingly ONLY in the abstract a minor amount of such information is given. It looks like the authors do not realize that an abstract is not an introduction nor a conclusion but a stand-alone text and that the conclusions in it have to be supported by information from the paper itself. Since this is missing, it is not clear at all what the real aspect of novelty is in this work.

Response 1: Thank you very much for your suggestion, We have made some changes in the manuscript.

Comment 2: On medical websites, I also see lower figures (like 8% overall) for the incidence of dysphagia. Please mention whether these data are for one certain country or region, or for the worldwide population.

Response 2: The worldwide population. We have changed the statement: Global data show that, dysphagia can occur at any age, but the risk is higher above the age of 65 years The prevalence is 13% in the age group 60-70 years, 16% in the age group 70-79 years, and about 33% in the group≥80 years.

Comment 3: L50 When an abbreviation such as EFSA is used, please inform the readers what this stands for. Same for the statement in lines 55-6which has no meaning without background knowledge.

Response 3: Thank you very much for your helpful suggestions. We've already explained the acronyms EFSA and others.

Comment 4: L66 Same for FSMP, which is written out in the abstract BUT the abstract is stand-alone and NOT an introduction, vide supra.

Response 4: Thank you very much for your suggestion. We've made additions.

Comment 5: Most important, the introduction should be extended according to the major issue mentioned above.

Response 5: Thank you very much for your suggestion, We have made some changes in the manuscript.

Comment 6: L114 combined with? what?

Response 6: Sorry, we made a mistake. It's starch, protein, and fats for mixing

Comment 7: The phrase pre-dextrinized starch is used several times in the paper but then I would expect to have a description of that in the ‘Materials’ section.

Response 7: Sorry, we made a mistake. It should be pre-gelatinised starch.

Comment 8: L173 Not any mass spectrometer that I ever met in the laboratory or that I see in the catalogues of lab suppliers can simulate swallowing behavior. Searching on the internet reveals something that is called a MASA test. Is there an unfortunate spelling error, do the authors mean another type of device, not a true mass spectrometer?

Response 8: I am very sorry that we made a mistake. We use textural analyzer. Thank you very much for your suggestion, and we have modified it in the manuscript.(Page 4, Line 192; Page 13, Line 476)

Comment 9: There are no numbers for DS in Table 1. OK, these are mentioned in paragraph 2.1 but that is far away from the results – why not mention the data here as well – much more comfortable for the reader.

Response 9: Thank you very much for your suggestion. We've made additions.

Comment 10: Chocking?? Must be choking probably?

Response 10:  Sorry, we made a mistake. It should be choking.

Comment 11: See the main general comment. The conclusions should be extended by an assessment of the novel versus the existing FMSP formulations. By the way, it looks like two times the same sentence with only one term different. This looks bad and can be formulated more logically.

Response 11: Thank you very much for your suggestion, We have made some changes in the manuscript. The new conclusion has been revised to read:

“The study investigated the feasibility of commercially available natural and pregelatinised starch starches from maize and cassava, acetylated and phosphorylated, as thickening agents for special medical food chyme. The rheological results showed that the chyme with thickening agents was a pseudoplastic fluid with good shear thinning ability, good thixotropic recovery, suitable viscoelasticity, and significant viscosity reduction in combination with protein, which is expected to be a new type of thickening agent for dysphagia. Pectin thickeners are commonly used in liquid beverages (water, juice, milk, etc.), but patients with dysphagia need whole-nutrient chyme to help them recover, and our thickeners provide an advantage for adding more energetic substances to swallowed food formulations. This study will provide a theoretical basis for the development of dysphagia foods.”

Reviewer 3 Report

Comments and Suggestions for Authors

This manuscript by Youdong Li et al. describes the characterisations of various starch-based additives for use as thickeners in foods suitable for people with difficulty in swallowing (dysphagia).  While the topic is important and merits investigation, I believe there are a significant number of mainly minor problems with the present manuscript that should be addressed before it could be considered as suitable for publication.

L35: '...the aggravation of population ageing...'  I think 'aggravation' is the wrong word.  (It suggests annoyance that the population is ageing.)  I suggest a better phrase would be:

'With the challenges posed by population ageing, there is a need...'

L50: Please define the abbreviation EFSA.  (It may not be known be readers outside the food science community.)

L55-56: 'Authorised for use as a food additive according to Regulation (EC) No 1333/2008.'  This phrase lacks a noun and is not a sentence.

L148-149:  The authors state that the rheometer was fitted with cone and plate geometry, with a gap of 1000 µm.  Cone and plate geometry is usually described in terms of its diameter, the opening angle and truncation distance (i.e. the height of the cone that is missing, to avoid a sharp point impinging on the rheometer plate.

A gap of 1000 µm sounds like a very large value for cone-and plate geometry.  What were the diameter and opening angle, please?

A gap of 1000 µm suggests a parallel plate geometry.  Can the authors check this, please.

L168: 0.34/mm/s (I think this should be 0.34 mm/s)

L173 and L403:  The text states that a 'mass spectrometer' was used.  I am unsure of what the authors mean by this.  The term 'mass spectrometer' usually refers to a device for analysing the mass-to-charge ratio of molecules or molecular fragments after they have been ionised, e.g. by collisions with high energy electrons.

I don't think that is what was being used here.  Can the authors check and provide a clearer description, please.

Was this a proprietary device?  If it was a commercial apparatus, the authors should also identify the manufacturer, please, which would also help the reader to understand the experimental procedure.

L187:  Please state the internal diameters of the syringe barrel and outlet tip.  These parameters could affect the draining rates in the experiments.

Fig. 1: The scale bars for the micrographs are difficult to see.  It would help the reader if the scale bars for each image were easier to see, please.

Fig. 2:  The FTIR spectra appear strange, with some unexpected features:

(i) The peaks towards the lower wavenumber end of the spectrum appear surprisingly stronger than those at higher wavenumber, including the O-H stretching bands, which are usually very strong in starch.  Can the authors confirm whether an ATR correction (to allow for the different penetration depths of the evanescent waves at different frequencies) was applied, please?

(ii) The CS spectra all show strong 'ripples' between 3500 and 4000 cm-1.  Can the authors explain what these are, please?  Are they some kind of artifact due to poor baseline subtraction?

(iii) The CS spectra also show strong negative peaks around 2300 cm-1.  This is likely to be due to there being more CO2 in the optical path during the background collection, compared with the sample scans.  Were the optical paths purged to remove atmospheric (or exhaled) CO2?  The authors should clarify this in the experimental method, please.

(iv) The CD-PCT and CD-ACT spectra show 'dips' around 2900 cm-1, where peaks due to C-H stretching should appear.  Might that indicate that the ATR crystal was contaminated (e.g. by hydrocarbons) when the corresponding background scans were collected?  Can the authors check this, please.

(v) In interpreting the spectra, the authors should be aware that the ATR-FTIR method samples the electromagnetic absorbance from a region up to a few micrometres from the surface of the  internal reflection element (the ATR crystal).  Hence, the quality of the spectra can depend on the granularity of the material being analysed and the clamping pressure used.

In view of the useful chemical information that could be obtained from ATR-FTIR, I suggest the authors should attempt to obtain better spectra and provide more detailed interpretations.

Fig. 3:  In part A, it is difficult to read the numbers on the axes and the x-axis labels.  Please ensure that sufficiently large font sizes are used - especially, for when the figures are reproduced in the final paper.

Can the authors explain what the labels a-f on the error bars in part B mean, please.

L293: The authors use the term 'pre-dextrinized starch'.  Is that the same as 'pre-gelatinized starch' used elsewhere (e.g. L63)?  If these terms are synonymous, that should be clarified where they first appear in the text.  Otherwise, it would be helpful to describe how they differ, please.

L325 The equation for the Ostwald-de-Waele model is not Eqn. 1.  An un-numbered equation for the volume of sample extruded appears between lines 169 and 170.

Note: that equation could also be simplified by taking out a factor of t.pi/4 from both sides.

The Ostwald-de-Waele power-law model (Eq. 1) should produce straight lines on double-logarithmic axes, whereas some of the data shown in Fig. 5 was distinctly curved.  It would be useful if the authors could show the models fitted to the data, please.

Also, why was the Ostwald-de-Waele model chosen?  Were any other models (e.g Herschel-Bulkley or Carreau-Yasuda) considered?  Can the authors comment, please.

Fig. 7: The numbers on the axes and the legends in these figures are difficult to read.  Hence, understanding the information conveyed in them is very difficult.  I suggest that the various components in Fig. 7 should be larger.   

The authors should that all of their figures can be read easily - especially considering their appearance in the final paper layout.

L381: 'chocking'  (choking).

L386: '...easily detach from the teeth/palate and suitable for... (...easily detach from the teeth/palate and are suitable for...)

Conclusions: It is not clear from the Conclusions, whether the authors are recommending one or more, or all of the formulations studied.  Can the authors clarify this, please.

Author Response

Reviewer #3: 

Comment 1: L35: '...the aggravation of population ageing...'  I think 'aggravation' is the wrong word.  (It suggests annoyance that the population is ageing.)  I suggest a better phrase would be:
'With the challenges posed by population ageing, there is a need...'

Response 1: Thank you very much for your suggestion, We have made some changes in the manuscript.(Page 1, Line 37)

Comment 2: L50: Please define the abbreviation EFSA.  (It may not be known be readers outside the food science community.)

Response 2: Thank you very much for your suggestion, We have made some changes in the manuscript.(Page 2, Line 54)

Comment 3: L55-56: 'Authorised for use as a food additive according to Regulation (EC) No 1333/2008.'  This phrase lacks a noun and is not a sentence.

Response 3: Thank you very much for your suggestion, We have made some changes in the manuscript.(Page 2, Line 59-61)

Comment 4: L148-149:  The authors state that the rheometer was fitted with cone and plate geometry, with a gap of 1000 µm.  Cone and plate geometry is usually described in terms of its diameter, the opening angle and truncation distance (i.e. the height of the cone that is missing, to avoid a sharp point impinging on the rheometer plate.
A gap of 1000 µm sounds like a very large value for cone-and plate geometry. What were the diameter and opening angle, please?
A gap of 1000 µm suggests a parallel plate geometry. Can the authors check this, please.

Response 4: Thank you very much for your suggestion, We have made some changes in the manuscript.(Page 4, Line 166)

Comment 5: L168: 0.34/mm/s (I think this should be 0.34 mm/s)

Response 5: Thank you very much for your suggestion, We have made some changes in the manuscript.(Page 4, Line 188)

Comment 6: L173 and L403:  The text states that a 'mass spectrometer' was used.  I am unsure of what the authors mean by this.  The term 'mass spectrometer' usually refers to a device for analysing the mass-to-charge ratio of molecules or molecular fragments after they have been ionised, e.g. by collisions with high energy electrons.
I don't think that is what was being used here.  Can the authors check and provide a clearer description, please.
Was this a proprietary device?  If it was a commercial apparatus, the authors should also identify the manufacturer, please, which would also help the reader to understand the experimental procedure.
Response 6: I am very sorry that we made a mistake. We use textural analyzer. Thank you very much for your suggestion, and we have modified it in the manuscript. (Page 4, Line 192; Page 13, Line 476)

Comment 7: L187:  Please state the internal diameters of the syringe barrel and outlet tip.  These parameters could affect the draining rates in the experiments.

Response 7: Thank you very much for your suggestion, we have added relevant information to the manuscript, and the new sentence is: “In this study, the 10 mL syringe experiment (the inner diameter of the syringe barrel and the outlet tip are 15mm and 0.7mm, respectively, predominantly used to measure drink thickness) was applied. A syringe was held vertically with its tip blocked. "(Page 5, Line 204-207)

Comment 8: Fig. 1: The scale bars for the micrographs are difficult to see.  It would help the reader if the scale bars for each image were easier to see, please.

Response 8: Thank you very much for your suggestion, we have sent the original image to the editor.

Comment 9: Fig. 2:  The FTIR spectra appear strange, with some unexpected features:
(i) The peaks towards the lower wavenumber end of the spectrum appear surprisingly stronger than those at higher wavenumber, including the O-H stretching bands, which are usually very strong in starch.  Can the authors confirm whether an ATR correction (to allow for the different penetration depths of the evanescent waves at different frequencies) was applied, please?

Response 9 (i) : We did a background scan with the air as the backdrop.
(ii) The CS spectra all show strong 'ripples' between 3500 and 4000 cm-1.  Can the authors explain what these are, please?  Are they some kind of artifact due to poor baseline subtraction?

Response 9 (ii): The samples absorb bands (corresponding to -OH groups) between 3500 and 400 cm-1 . The amplitude may be due to more precise binding of internal and intermolecular hydrogens.
(iii) The CS spectra also show strong negative peaks around 2300 cm-1.  This is likely to be due to there being more CO2 in the optical path during the background collection, compared with the sample scans.  Were the optical paths purged to remove atmospheric (or exhaled) CO2?  The authors should clarify this in the experimental method, please.

Response 9 (iii): I am very sorry that we made a mistake. We don't seem to be removing CO2 from the atmosphere.

(iv) The CD-PCT and CD-ACT spectra show 'dips' around 2900 cm-1, where peaks due to C-H stretching should appear.  Might that indicate that the ATR crystal was contaminated (e.g. by hydrocarbons) when the corresponding background scans were collected?  Can the authors check this, please.

Response 9 (iv) :It has been verified that there is no contamination and that the CD-PCT and CD-ACT spectra show a ‘dip’ around 2900 cm-1 due to the introduction of new phosphate groups and acetyl groups by phosphorylation and acetylation, resulting in a change in the C-H stretching peak.
(v) In interpreting the spectra, the authors should be aware that the ATR-FTIR method samples the electromagnetic absorbance from a region up to a few micrometres from the surface of the  internal reflection element (the ATR crystal).  Hence, the quality of the spectra can depend on the granularity of the material being analysed and the clamping pressure used. In view of the useful chemical information that could be obtained from ATR-FTIR, I suggest the authors should attempt to obtain better spectra and provide more detailed interpretations.

Response 9 (v) :Thank you very much for your suggestion.

Comment 10: Fig. 3:  In part A, it is difficult to read the numbers on the axes and the x-axis labels.  Please ensure that sufficiently large font sizes are used - especially, for when the figures are reproduced in the final paper. Can the authors explain what the labels a-f on the error bars in part B mean, please.

Response 10: Thank you very much for your suggestion, we have sent the original image to the editor. An explanation of the letters is also added to the illustration. (Page 7, Line 286)

Comment 11: L293: The authors use the term 'pre-dextrinized starch'. Is that the same as 'pre-gelatinized starch' used elsewhere (e.g. L63)? If these terms are synonymous, that should be clarified where they first appear in the text. Otherwise, it would be helpful to describe how they differ, please.

Response 11: Thank you very much for pointing out our mistake, it should be "pre-gelatinized", we have corrected it in the manuscript. (Page 8, Line 324)

Comment 12: L325 The equation for the Ostwald-de-Waele model is not Eqn. 1. An un-numbered equation for the volume of sample extruded appears between lines 169 and 170.

Note: that equation could also be simplified by taking out a factor of t.pi/4 from both sides.

The Ostwald-de-Waele power-law model (Eq.1) should produce straight lines on double-logarithmic axes, whereas some of the data shown in Fig. 5 was distinctly curved. It would be useful if the authors could show the models fitted to the data, please.

Also, why was the Ostwald-de-Waele model chosen? Were any other models (e.g Herschel-Bulkley or Carreau-Yasuda) considered? Can the authors comment, please.

Response 12: Thank you so much for this comment. Common non-Newtonian fluids are Ostwald-de-Waele model, Herschel-Bulkley and Carreau-Yasuda. The Ostwald-de-Waele model is suitable for pseudoplastic or dilatant fluids with a wide range of shear deformation rates. Because of its simplicity in formula, it has great practical value in engineering. Therefore, we chose Ostwald-de-Waele model in this study.

Comment 13: Fig. 7: The numbers on the axes and the legends in these figures are difficult to read. Hence, understanding the information conveyed in them is very difficult. I suggest that the various components in Fig. 7 should be larger. The authors should that all of their figures can be read easily - especially considering their appearance in the final paper layout.

Response 13: We are very sorry for the low quality of the graphics, we have sent the original graphics to the editor, thank you for this comment.

Comment 14: L381: 'chocking'  (choking).

Response 14: we are sorry that we made a mistake. We have corrected it in the manuscript.

Comment 15: L386: '...easily detach from the teeth/palate and suitable for... (...easily detach from the teeth/palate and are suitable for...)

Response 15: Thank you very much for your suggestion, we have revised it in the manuscript. (Page 13, Line 453)

Comment 16: Conclusions: It is not clear from the Conclusions, whether the authors are recommending one or more, or all of the formulations studied.  Can the authors clarify this, please.

Response 16: Thank you very much for this comment. We have evaluated the effects of different modified starches on whole-nutrient chyme and found that they are suitable for patients with dysphagia and have better effects than pectin thickener (they are suitable for fluids). Therefore, we recommend the application of multiple modified starches.

Round 2

Reviewer 2 Report

Comments and Suggestions for Authors

I am pleased to see that there were many revisions and additions made by the authors of this manuscript, according to the comments of the referees and the editor. Overall, we can be satisfied to a large degree with the revised manuscript. Still, I think that the distinction between currently applied products for dysphagia patients and the newly developed materials can be made sharper. Yes, it can be found in the text, but it is not so easy to recognize. So, this could still be emphasized more clearly. For example, in the conclusion, it reads that pectin is used a lot – but is that in general, for products not targeted for dysphagia patients, or are pectin-based chyme’s already on the market for this application? I would recommend to also mention such in the introduction paragraph, perhaps in the text of lines 67-8. There, it is not clear whether the commercial starch derivatives as mentioned are already applied in diet food for dysphagia patients or that such is new. Or is the novelty of this work that existing formulations are investigated much deeper? This should be clarified better.

Other comments have been answered satisfactorily. Several errors ranging from typing errors until mentioning the wrong type of equipment for swallowing simulation have been corrected.

So, with a sharper definition of the real novelty in this work, the paper can be accepted with minor further improvements.

Author Response

Thank you very much for your suggestion, We have made some changes in the manuscript. The new abstract has been revised to read:

A dysphagia diet is a special dietary programme. The development and design of foods for dysphagia should consider both swallowing safety and food nutritional quality. In this study, we investigated the rheological properties (viscosity, thixotropy and viscoelasticity), textural properties and swallowing behaviour of commercially available natural, pregelatinised, acetylated and phosphorylated maize starch and tapioca starch. The results showed that all the samples belonged to food grade 3 in the framework of the International Dysphagia Dietary Standardization Initiative (IDDSI) and exhibited shear-thinning behaviour in favour of dysphagia patients, except for the sample containing pregelatinised starch, which was grade 2. Rheological tests showed that so the samples had good structural recovery properties. At the same starch concentration, the elastic modulus of phosphorylated cassava starch FSMP was significantly greater than that of the starch solution, whereas that of acetylated starch was significantly less than that of the starch solution, and the combination of acetylated starch and protein led to a significant viscosity reduction phenomenon, resulting in FSMPs with good stability and fluidity; this may provide an opportunity for the incorporation of more high-energy substructures. The textural results showed that all the samples possessed textural properties of low hardness, low adhesion and high cohesion, all of which could be used as food for dysphagia patients. This study may provide a theoretical basis for the creation and design of novel nutritional foods for dysphagia.